# The Content Specificity and Generality of the Relationship between Mathematical Problem Solving and Affective Factors

Yuno Shimizu 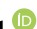

Center for Curriculum Redesign, Hyogo University of Teacher Education, Kato 673-1494, Hyogo Prefecture, Japan; yshimizu@hyogo-u.ac.jp

**Abstract:** Research has revealed that both cognitive factors, such as knowledge, problem solving strategies and affective factors, such as motivation and emotions, strongly influence mathematical problem solving. However, few studies have examined the content specificity and generality of the relationship between mathematical problem solving and affective factors. This study examines the content specificity and generality of the relationship between mathematical problem solving, task value, math anxiety and engagement among high school students. Japanese second-year high school students ($n = 240$) completed questionnaires. The multilevel structural equation modelling revealed that utility value for entrance examinations and emotional engagement positively affected mathematical problem solving via cognitive engagement between various contents level. Emotional engagement positively affected mathematical problem solving via cognitive engagement within a certain content level. The results suggest that promoting the perception that learning mathematics has high utility value for university entrance examinations across various contents can increase students' cognitive engagement and, therefore, improve mathematical problem solving. Furthermore, both increasing students' emotional engagement only when they learn certain content and consistently increasing it may improve cognitive engagement and, therefore, allow learners to better solve mathematical problems. The study's findings have significant implications for educational practice and mathematical problem-solving research.

**Keywords:** content specificity; content generality; task value; math anxiety; engagement; mathematical problem solving

## 1. Introduction

Today, the importance of science, technology, engineering and mathematics (STEM) education is increasing worldwide. As the language of science, mathematics within STEM education represents and interprets various phenomena in everyday life and scientific activities; therefore, improving problem solving skills in mathematics is an objective in both primary and secondary education [1]. In particular, the ability to solve problems in high school mathematics is essential not only for understanding events in everyday life and for professional education at universities but also for predicting future career prospects and income [2]. As such, educators must understand the determinants of problem solving in high school mathematics.

Previous research has indicated that both cognitive factors, such as knowledge, and problem solving strategies and affective factors, such as motivation and emotion, strongly influence mathematical problem solving [3,4]. For example, mathematics problem solving is associated with task value [5], math anxiety [6], self-efficacy [7], achievement goals [8] and intrinsic motivation [9]. Furthermore, the autonomous learning behavior model has captured the process of relating affective factors to mathematical problem solving [10], in which these factors influence mathematical problem solving through autonomous learning behavior. Empirical studies have also shown that affective factors influence mathematical

problem solving through learning engagement [11], challenge seeking for math [8], and learning approaches [12].

As shown above, previous studies have revealed the relationship between mathematical problem solving and affective factors and their processes. However, few researchers have addressed the question of the content specificity and generality of the relationship between mathematical problem solving and affective factors. Schukajlow et al. [13] highlighted that affective factors are content specific because students have different interests in math problems that are related to—or unrelated to—real situations. In addition, a meta-analysis of mathematical anxiety by Barroso et al. [6] showed a negative association between mathematical anxiety and mathematical problem solving, although the degree of this association varied according to the problem's content, for example, algebra or geometry problems. These findings suggest that the relationship between mathematical problem solving and affective factors and its process have both content-specific and general aspects.

The present study examines the content specificity and generality of the relationship between mathematical problem solving and affective factors, focusing on task value and mathematics anxiety as affective factors and engagement as a mediating factor. By identifying the effect of the content specificity and generality of this relationship, it may be possible to separate the common factors that should be considered across multiple types of content from those that should be considered only in specific content. Thus, identifying the importance of the content specificity and generality of this relationship has educational and academic significance because it can provide more detailed basic information to improve learners' mathematical problem solving than has been offered by the findings of previous studies.

### 1.1. Task Value and Mathematical Problem Solving

Task value indicates the subjective desirability of doing a task [14] and comprises three aspects: interest value, attainment value and utility value [15]. Interest value represents the trait enjoyment or fun of doing a task [16] and is similar to intrinsic motivation [17]. Attainment value is the importance of performing a given task well and is closely related to identity [15]. Finally, utility value is the importance of completing a task for future planning, for example, a task's usefulness for a future career, and is similar to extrinsic motivation [16] because the activity itself is not the goal [15]. In Eccles' expected value theory [17], these three task values are considered the determinants of achievement-related choices, behaviors and performance [15]. Furthermore, empirical studies have shown that interest value, attainment value, and utility value are positively associated with mathematical problem solving [5,12,18]. Since task value is an affective factor that promotes mathematical problem solving, it is a central focus of this study.

### 1.2. Math Anxiety and Mathematical Problem Solving

Math anxiety is defined as feelings of tension and anxiety that interfere with number operations and mathematical problem solving in a wide range of everyday life and academic situations [19]. Mathematics anxiety is a type of anxiety that emerges in mathematics-related situations and is a construct that is distinct from general anxiety and test anxiety [20]. Numerous studies have shown that math anxiety is a factor in the avoidance of mathematics in daily life and school and hinders learners' mathematical problem solving [6,19]. This study focuses on math anxiety since it can be defined as an affective factor that inhibits mathematical problem solving.

Math anxiety may influence mathematical problem solving by mediating learning engagement. The control-value theory [21] assumes that activating negative emotions, such as math anxiety, can cause a loss of cognitive resources by triggering thoughts that are unrelated to learning and achievement, leading to the avoidance of learning and lower achievement. Empirical studies have demonstrated that students with higher math anxiety are more likely to avoid learning math, such as the extent of high school math, the number of high schools' math courses, the intention to enroll in more math courses [20,22], and not using sophisticated problem solving strategies [23]. In other words, high levels of math

anxiety may inhibit learners' mathematical problem solving due to the avoidance of math learning engagement.

### 1.3. Engagement and Mathematical Problem Solving

Learning engagement is defined as active involvement in learning activities and is a multidimensional construct that includes behavioral, emotional and cognitive aspects [24,25]. Behavioral engagement refers to students' involvement in learning, comprising facets such as attitude, effort and persistence. Emotional engagement refers to state positive emotional responses during learning, such as learners' interest in, enjoyment of, and enthusiasm for learning. Lastly, cognitive engagement refers to cognitive participation, such as planning and monitoring learning and using learning strategies. In general, engagement is generally associated with desirable academic, social, and emotional outcomes [25], and similar results have been observed in mathematics education [11,26,27]. This study focuses on engagement given that it is considered a factor that can promote mathematical problem solving.

Engagement is assumed to be directly related to mathematical problem solving. Pekrun and Linnenbrink-Garcia [28] argued that engagement directly promotes achievement in control-value theory. Empirical studies have also found that engagement directly predicts mathematics achievement [11,26,27]. In particular, Fung et al. [26] and Shimizu [27] noted that cognitive engagement is strongly associated with achievement because it relates to various aspects of learning and is an indicator of learning motivation [26]. Thus, it is possible that high engagement, especially cognitive engagement, directly promotes mathematical problem solving.

### 1.4. Study Aims

This study examines the content specificity and generality of the relationship between mathematical problem solving, task value, math anxiety and engagement among high school students. The study clarifies the content specificity and generality of a hypothetical model (Figure 1), based on previous studies [5,6,11,12,18,21,23,26–28].

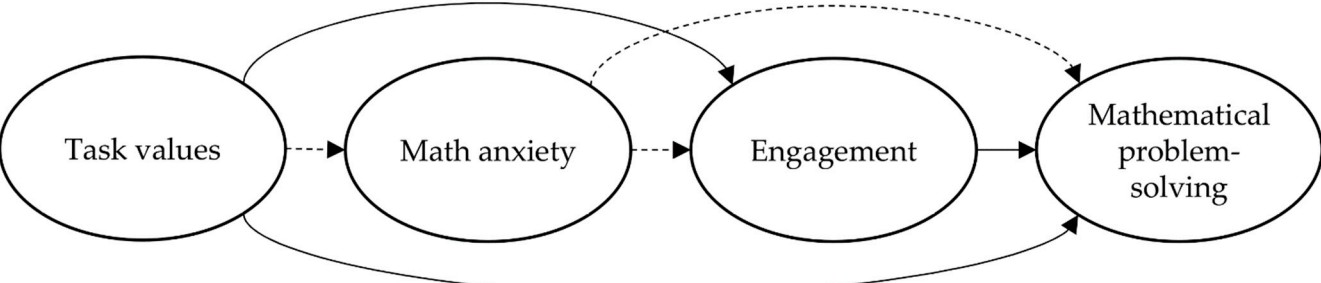

**Figure 1.** Hypothesis model of the current study. Solid lines represent positive relations; dashed lines represent negative relations.

The hypothetical model comprises the following five hypotheses:

**Hypothesis 1**: *task value is negatively related to math anxiety.*

**Hypothesis 2**: *math anxiety is negatively related to engagement.*

**Hypothesis 3**: *engagement is positively related to mathematical problem solving.*

**Hypothesis 4**: *task value is indirectly positively associated with mathematical problem solving through math anxiety and engagement.*

**Hypothesis 5**: *math anxiety is indirectly negatively related to mathematical problem solving through engagement.*

In addition, the study assumed that emotional engagement is positively associated with behavioral and cognitive engagement because emotions have motivational aspects that direct behavior and cognition [29].

## 2. Methods

### 2.1. Participants and Procedure

The study's participants included 240 second-year high school students (53% females, 47% males) enrolled in School A in the Tokyo area. School A is a preparatory school, and most students attend university after graduation. All participants were ethnic Japanese. Although the study did not collect age data on the participants, second-year high school students in Japan are typically 16–17 years old.

After requesting surveys from several high schools in the Tokyo area, permission was obtained from School A to conduct the survey as part of an improvement of math education practice in School A. Prior to the survey, the author explained the study's purpose to School A's teacher who was responsible for managing the math department and requested his cooperation. After obtaining the teacher's consent, the author sent a simultaneous email to all students regarding the purpose and content of the survey to recruit participants. As a result, 240 students cooperated with the survey.

The survey comprised two questionnaires. The first was administered during self-study time in math class one week prior to the first end-of-term examination in School A in September 2020 and measured task value, math anxiety and engagement. The second questionnaire survey was administered during self-study time in math class one week before the second end-of-term examination in School A in December 2020 and also measured task value, math anxiety and engagement.

At the time of the survey, the author explained to the participants both orally and via a clear statement on the questionnaire that (1) the responses to the survey were voluntary and unrelated to the participants' mathematics achievement, (2) participants' privacy would be protected because the responses to the survey would be processed statistically, and (3) the author would be responsible for disposing of the questionnaires.

Due to illness and other reasons, the students were absent from the classes where the survey was conducted, so data could not be obtained for 22 of the 240 participants in the first and 2 in the second survey. In addition, due to absence from the end-of-semester exam, data on mathematical problem solving at the first term-end could not be obtained from one participant.

### 2.2. Measures

The author and two mathematics teachers from junior high and high schools confirmed whether items of task value, math anxiety and engagement were appropriate for high school students and corrected the difficult expressions to understand. All items of task value, math anxiety and engagement have been presented in Appendix A.

#### 2.2.1. Task Value

Twelve items were used to measure task value, which were adapted from a mathematics-focused adaptation of the Japanese version of the task value scale [30] based on Eccles and Wigfield's original task value scale [31]. It is confirmed that the Japanese version of the task value scale [30] satisfies adequate convergent and discriminant validity and reliability. Incidentally, this study excluded one item from the Japanese version of the task value scale [30] that was a reversal item. The scale includes the following four subscales: interest value ($n = 3$), attainment value ($n = 3$), and two utility values for math practice's general value ($n = 3$) and its value for passing Japan's university entrance examinations ($n = 3$). Students rated the items on a 6-point Likert scale ranging from 'completely disagree' (1) to 'completely agree' (6).

### 2.2.2. Math Anxiety

Eight items were used to measure math anxiety. These items were developed based on items with large factor loadings in the Japanese mathematics anxiety rating scale [32], which was based on Richardson and Suinn's work [33]. The measure assessed students' math learning anxiety ($n = 4$) and math evaluation anxiety ($n = 4$). Students rated the items on a 6-point Likert scale ranging from 'not very anxious' (1) to 'very anxious' (6).

### 2.2.3. Engagement

Nine items were used to measure engagement in learning mathematics. The items related to behavioral and emotional engagement were based on Skinner et al.'s [34] scale, while those that measured cognitive engagement were based on items from Reeve and Tseng's [35] scale, which emphasizes semantic understanding. The scale included the following three subscales: behavioral engagement ($n = 3$), emotional engagement ($n = 3$) and cognitive engagement ($n = 3$). While Skinner et al.'s [34] scale was a 4-point Likert scale and Reeve and Tseng's [35] scale was a 7-point Likert scale, students rated the items on a 6-point Likert scale ranging from 'completely disagree' (1) to 'completely agree' (6) to be consistent with other measures in this study.

### 2.2.4. Mathematical Problem Solving

The scores of School A's first and second end-of-term exams were used to measure mathematical problem solving. Each end-of-term exam was prepared by the teacher in charge of the participants' class at School A, based on the authorized textbooks and the course of study for mathematics in high school in Japan [1]. The proportion of problems in the first and second end-of-term exams was 70% from basic examples and exercises from the textbook with different values and conditions and 30% from problems applying basic content. The first end-of-term exam contained 22 calculation and proof problems on plane vector representations and operations. The second end-of-term exam contained 21 calculation and proof problems on space vector representations and operations and operations on matrices. The scores of the first and second examinations ranged from 0 to 100.

### 2.3. Data Analysis

The data analysis comprised three main steps. First, a preliminary analysis was conducted to check the scale's validity and calculate descriptive statistics. The structure of the scale was confirmed by confirmatory factor analysis (restricted maximum likelihood method). The scale's reliability was examined using the $\omega$ coefficient with the number of group factors set to 1 and composite reliability (*CR*). The average variance extracted (*AVE*) was calculated to verify the convergent validity. In general, it is recommended that the *AVE* value is 0.50 or over [36]. However, even if the *AVE* value is 0.50 or under, the scale's validity is sufficient if the *CR* value is 0.60 or over [36]. The discriminant validity was also tested by comparing the square root of the *AVE*s with the correlation coefficient of the scale. Regarding the discriminant validity, it is recommended that the square root of the *AVE*s is greater than or equal to the correlation coefficient between each scale [37]. Second, the intraclass correlation coefficient (*ICC*) was calculated to determine the degree of similarity between task value, math anxiety, engagement, and mathematical problem solving at the end of the first and second terms. In this study, the *ICC* was obtained by dividing the variance between the first and second end-of-term scores for the variable concerned by the overall variance. Therefore, the *ICC* in this study represents the component shared between the first and second end-of-term scores for the variable concerned. Third, multilevel structural equation modelling was conducted based on the hypothetical model shown in Figure 1 to examine the content specificity and generality of the relationship between mathematical problem solving, task value, math anxiety and engagement. Based on Muthén and Muthén [38], this study used restricted maximum likelihood method, which has less bias for relatively small sample sizes. Multilevel structural equation modelling is a method that decomposes data into 'within models', which are unique components within primary

extraction units, and 'between models', which are shared components between primary extraction units, and then models each independently [38]. This study compiled a two-level hierarchical data structure with task value, engagement, and mathematical problem solving at different mathematical content learning time points (first end-of-term, second end-of-term) nested within individuals. In other words, the primary extraction unit in this study is each point in time in which participants learnt each of the different mathematical contents. Thus, the within model in this study represents a unique component for each mathematical content of the relationship between mathematical problem solving, task value, math anxiety, and engagement. For example, if a significant negative path from math anxiety to mathematical problem solving is found in the within model, the higher the math anxiety in a certain content, the less able students are to solve the mathematical problem in that content. On the other hand, the between model in this study represents a common component across various mathematical contents of the relationship between mathematical problem solving, task value, math anxiety, and engagement. For example, if a significant negative path from math anxiety to mathematical problem solving is found in the between model, the higher the math anxiety across various contents, the less able students are to solve mathematical problems in various contents. In this study, multilevel structural equation modelling was performed in Mplus (version 8), and other data analysis was performed in R4.1.1, an open software environment. For the above analyses, missing values were dealt with using the listwise method.

## 3. Results

### 3.1. Preliminary Analysis

The results of a confirmatory factor analysis of the assumed scale structures for task value, math anxiety and engagement are presented in Table 1. All goodness-of-fit indicators were good for task value and engagement. Math anxiety had a poor RMSEA, but the other goodness-of-fit indicators were good. Therefore, the assumed scale structure of task value, math anxiety and engagement was valid.

**Table 1.** Results of confirmatory factor analysis.

| | First Term-End | | | | Second Term-End | | | |
|---|---|---|---|---|---|---|---|---|
| | *CFI* | *TLI* | *RMSEA* | *SRMR* | *CFI* | *TLI* | *RMSEA* | *SRMR* |
| Task value | 0.94 | 0.91 | 0.09 | 0.07 | 0.97 | 0.96 | 0.06 | 0.06 |
| Math anxiety | 0.95 | 0.92 | 0.13 | 0.05 | 0.95 | 0.92 | 0.12 | 0.04 |
| Engagement | 0.98 | 0.97 | 0.07 | 0.04 | 0.97 | 0.96 | 0.07 | 0.05 |

The $\omega$ coefficients, *CR*, *AVE* and descriptive statistics are shown in Table 2. At the first and second term-end, the means of the task value, math evaluation anxiety, and engagement scores were also above the six-point Likert scale semantic median of 3.50, and the means of the respondents' math learning anxiety scores were below the six-point Likert scale semantic median of 3.50. The reliability of each scale was confirmed to be good because the value of the $\omega$ coefficient and *CR* was greater than 0.62. Regarding the convergent validity, the *AVE* values of utility value for the entrance examinations at the first term-end, attainment value at the first and second term-end, and cognitive engagement at the second term-end were less than 0.50, while the *CR* values of these scales were greater than 0.60. In addition, the *AVE* values for these other scales were greater than 0.50. Thus, scales of task value, math anxiety, and engagement had adequate convergent validity.

**Table 2.** Descriptive statistics for task value, math anxiety, engagement and mathematical problem solving.

| | First Term-End | | | | | | Second Term-End | | | | | |
|---|---|---|---|---|---|---|---|---|---|---|---|---|
| | N | ω | CR | AVE | M | SD | N | ω | CR | AVE | M | SD |
| 1. Utility value for practice | 218 | 0.88 | 0.88 | 0.70 | 3.81 | 1.11 | 237 | 0.85 | 0.85 | 0.65 | 3.89 | 1.05 |
| 2. Utility value for the entrance examinations | 218 | 0.72 | 0.72 | 0.49 | 4.64 | 0.93 | 237 | 0.74 | 0.75 | 0.52 | 4.67 | 0.98 |
| 3. Interest value | 218 | 0.86 | 0.86 | 0.69 | 3.79 | 1.15 | 236 | 0.91 | 0.92 | 0.79 | 3.83 | 1.24 |
| 4. Attainment value | 218 | 0.66 | 0.62 | 0.40 | 4.20 | 0.97 | 234 | 0.68 | 0.66 | 0.46 | 4.24 | 0.98 |
| 5. Math learning anxiety | 218 | 0.88 | 0.88 | 0.64 | 3.19 | 1.26 | 238 | 0.87 | 0.87 | 0.62 | 3.04 | 1.23 |
| 6. Math evaluation anxiety | 218 | 0.92 | 0.92 | 0.74 | 4.77 | 1.34 | 235 | 0.92 | 0.92 | 0.75 | 4.78 | 1.26 |
| 7. Behavioral engagement | 218 | 0.85 | 0.84 | 0.65 | 5.14 | 0.73 | 237 | 0.86 | 0.86 | 0.68 | 4.67 | 0.99 |
| 8. Emotional engagement | 218 | 0.92 | 0.92 | 0.80 | 4.17 | 1.12 | 237 | 0.89 | 0.90 | 0.74 | 3.79 | 1.12 |
| 9. Cognitive engagement | 218 | 0.85 | 0.85 | 0.66 | 4.90 | 0.86 | 238 | 0.72 | 0.72 | 0.46 | 4.56 | 0.86 |
| 10. Mathematical problem solving | 239 | - | - | - | 58.73 | 19.03 | 240 | - | - | - | 45.79 | 16.79 |

Table 3 shows the Pearson correlation matrix with the square root of the *AVE*s. The square root of the *AVE*s was greater than or equal to the correlation coefficient between each scale. Thus, scales of task value, math anxiety and engagement had adequate discriminant validity. In the following analysis of this study, the arithmetic mean of each item was used as the score for the task value, math anxiety and engagement subscales.

**Table 3.** Pearson correlation matrix for task value, math anxiety, engagement and mathematical problem solving.

| | 1 | 2 | 3 | 4 | 5 | 6 | 7 | 8 | 9 | 10 |
|---|---|---|---|---|---|---|---|---|---|---|
| 1. Utility value for practice | (0.84/0.80) | 0.49 *** | 0.57 *** | 0.50 *** | −0.29 *** | −0.23 *** | 0.05 | 0.40 *** | 0.19 ** | 0.30 *** |
| 2. Utility value for entrance examinations | 0.59 *** | (0.70/0.72) | 0.40 *** | 0.60 *** | −0.27 *** | −0.12 | 0.15 * | 0.28 *** | 0.16 * | 0.26 *** |
| 3. Interest value | 0.49 *** | 0.40 *** | (0.83/0.89) | 0.45 *** | −0.53 *** | −0.43 *** | 0.11 | 0.69 *** | 0.33 *** | 0.37 *** |
| 4. Attainment value | 0.63 *** | 0.62 *** | 0.34 *** | (0.63/0.67) | −0.14 * | −0.06 | 0.16 * | 0.31 *** | 0.18 ** | 0.16 * |
| 5. Math learning anxiety | −0.19 ** | −0.20 ** | −0.47 *** | −0.07 | (0.80/0.79) | 0.67 *** | −0.07 | −0.35 *** | −0.21 *** | −0.36 *** |
| 6. Math evaluation anxiety | −0.23 *** | −0.15 * | −0.39 *** | −0.06 | 0.66 *** | (0.80/0.79) | −0.05 | −0.30 *** | −0.11 | −0.28 *** |
| 7. Behavioral engagement | 0.24 *** | 0.31 *** | 0.24 *** | 0.22 ** | −0.09 | −0.04 | (0.81/0.83) | 0.35 *** | 0.48 *** | 0.16* |
| 8. Emotional engagement | 0.40 *** | 0.36 *** | 0.70 *** | 0.27 *** | −0.36 *** | −0.26 *** | 0.50 *** | (0.90/0.86) | 0.49 *** | 0.25 *** |
| 9. Cognitive engagement | 0.35 *** | 0.38 *** | 0.41 *** | 0.30 *** | −0.25 *** | −0.16 * | 0.68 *** | 0.68 *** | (0.81/0.68) | 0.34 *** |
| 10. Mathematical problem−solving | 0.11 | 0.18 ** | 0.34 *** | 0.14 * | −0.29 *** | −0.33 *** | 0.23 ** | 0.33 *** | 0.32 *** | - |

Pearson correlations for first term-end are displayed below the diagonal; Pearson correlations for second term-end are displayed above the diagonal. The values on the diagonal are the square root of *AVE*. The value on the left is the value of the square root of the *AVE* at the first term-end, and the value on the right is the value of the square root of the *AVE* at the second term-end. *** $p < 0.001$, ** $p < 0.01$, * $p < 0.05$.

In addition, at the end of both the first and second terms, mathematical problem solving was positively correlated with task value and engagement and negatively correlated with math anxiety.

*3.2. ICCs*

The *ICC*s for task value and math anxiety, engagement and mathematical problem solving are shown in Table 4. The *ICC*s were greater than 0.57 for task value, 0.67 for math anxiety, 0.36 for engagement and 0.44 for mathematical problem solving. The largest *ICC* was 0.78 for interest value and the smallest was 0.36 for behavioral engagement.

**Table 4.** *ICC*s for task value, math anxiety, engagement and mathematical problem solving.

| | ICC | 95%CI |
|---|---|---|
| 1. Utility value for practice | 0.70 | (0.64, 0.77) |
| 2. Utility value for entrance examinations | 0.57 | (0.49, 0.66) |
| 3. Interest value | 0.78 | (0.73, 0.83) |
| 4. Attainment value | 0.65 | (0.57, 0.72) |
| 5. Math learning anxiety | 0.67 | (0.59, 0.74) |
| 6. Math evaluation anxiety | 0.71 | (0.64, 0.78) |
| 7. Behavioral engagement | 0.36 | (0.24, 0.47) |
| 8. Emotional engagement | 0.57 | (0.48, 0.66) |
| 9. Cognitive engagement | 0.40 | (0.29, 0.51) |
| 10. Mathematical problem solving | 0.44 | (0.33, 0.54) |

### 3.3. Multilevel Structural Equation Modelling

The goodness-of-fit indices for the hypothetical model (Figure 1) were good (*CFI* = 0.99, *TLI* = 0.95, *RMSEA* = 0.04, *SRMR* (within) = 0.03, *SRMR* (between) = 0.04, *AIC* = 13888.72, and *BIC* = 14339.94). However, the analysis was conducted while deleting these paths as there were some insignificant paths. The results of the re-analysis are shown in Figure 2. The goodness-of-fit indices for the model (Figure 2) were good (*CFI* = 1.00, *TLI* = 1.00, *RMSEA* = 0.00, *SRMR* (within) = 0.04, *SRMR* (between) = 0.07, *AIC* = 13838.29, and *BIC* = 14134.92). Moreover, the information criteria for the model in Figure 2 was better than that for the hypothetical model (Figure 1). Therefore, the model in Figure 2 was accepted.

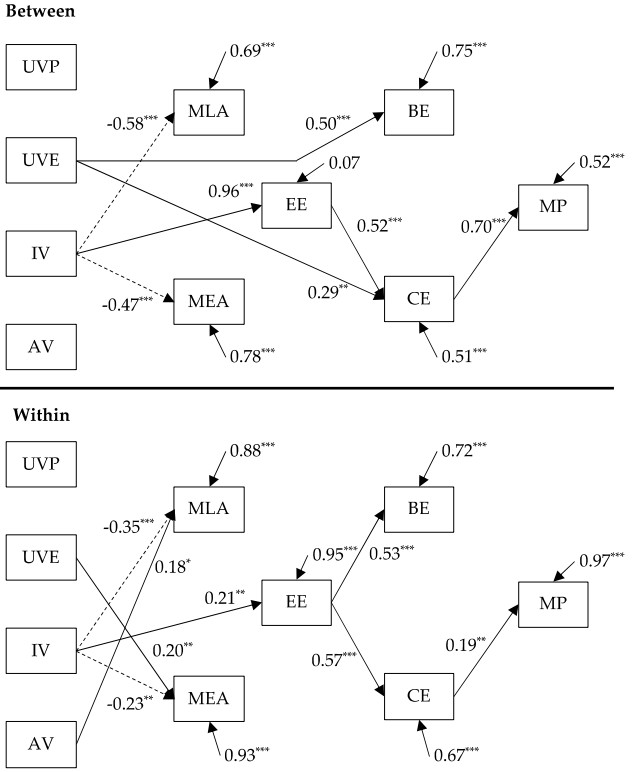

**Figure 2.** Results of multilevel structural equation modelling. UVP, utility value for practice; UVE, utility value for the entrance examinations; IV, interest value; AV, attainment value; MLA, math learning anxiety; MEA, math evaluation anxiety; EE, emotional engagement; BE, behavioral engagement; CE, cognitive engagement; and MP, mathematical problem solving. The values in the figure are standardized path coefficients and error variance. In particular, the values of the standardized error variance in the between model represent the random intercept. Solid lines represent positive relations; dashed lines represent negative relations. *** *p* < 0.001, ** *p* < 0.01, * *p* < 0.05.

The results for the within model shown in Figure 2 revealed that interest value was negatively related to math learning anxiety and math evaluation anxiety (β = −0.35 and −0.23, respectively) and positively related to emotional engagement (β = 0.21). Attainment value was positively related to math learning anxiety (β = 0.20), while utility value for entrance examinations was positively related to math evaluation anxiety (β = 0.20). Emotional engagement was positively related to behavioral and cognitive engagement (β = 0.53 and 0.57, respectively), and cognitive engagement was positively related to mathematical problem solving (β = 0.19). The within model explained 12% of the variance in math learning anxiety, 7% of the variance in math evaluation anxiety, 5% of the variance in emotional engagement, 28% of the variance in behavioral engagement, 33% of the variance in cognitive engagement and 3% of the variance in mathematical problem solving.

The results for the between model shown in Figure 2 revealed that interest value was negatively related to math learning anxiety and math evaluation anxiety (β = −0.58 and −0.47, respectively) and positively related to emotional engagement (β = 0.96). Utility value for entrance examinations was positively related to behavioral and cognitive engagement (β = 0.50 and 0.29, respectively). Meanwhile, emotional engagement was positively related to cognitive engagement (β = 0.52), and cognitive engagement was positively related to mathematical problem solving (β = 0.70). The within model explained 33% of the variance in math learning anxiety, 22% of the variance in math evaluation anxiety, 93% of the variance in emotional engagement, 25% of the variance in behavioral engagement, 49% of the variance in cognitive engagement, and 48% of the variance in mathematical problem solving.

Since it was assumed that interest value, utility value for entrance examinations, and emotional engagement had indirect effects on mathematical problem solving, a mediation analysis was conducted (Table 5). The results for the within model shown in Table 5 revealed that interest value had a positive indirect effect on mathematical problem solving via emotional and cognitive engagement (β = 0.02). Emotional engagement had a positive indirect effect on mathematical problem solving via cognitive engagement (β = 0.11). The results for the between model (Table 5) revealed that utility value for entrance examinations had a positive effect on mathematical problem solving via cognitive engagement (β = 0.20), interest value had a positive effect on mathematical problem solving via emotional and cognitive engagement (β = 0.35), and emotional engagement had a positive effect on mathematical problem solving via cognitive engagement (β = 0.36).

**Table 5.** Indirect effects.

|  | β | 95%CI | SE |
|---|---|---|---|
| **Within** | | | |
| Interest value | | | |
| →Emotional engagement | 0.02 * | (0.00, 0.04) | 0.01 |
| →Cognitive engagement | | | |
| →Mathematical problem solving | | | |
| Emotional engagement | | | |
| →Cognitive engagement | 0.11 ** | (0.03, 0.18) | 0.04 |
| →Mathematical problem solving | | | |
| **Between** | | | |
| Utility value for entrance examinations | | | |
| →Cognitive engagement | 0.20 ** | (0.08, 0.33) | 0.06 |
| →Mathematical problem solving | | | |
| Interest value | | | |
| →Emotional engagement | 0.35 *** | (0.18, 0.52) | 0.09 |
| →Cognitive engagement | | | |
| →Mathematical problem solving | | | |
| Emotional engagement | | | |
| →Cognitive engagement | 0.36 *** | (0.19, 0.53) | 0.09 |
| →Mathematical problem solving | | | |

*** $p < 0.001$, ** $p < 0.01$, * $p < 0.05$.

## 4. Discussion

This study examined the content specificity and generality of the relationship between mathematical problem solving, task value, math anxiety and engagement among high school students. In the following sections, the multilevel structural equation modelling results are discussed in terms of the contribution of task value, mathematics anxiety and engagement to the content specificity and generality of the relationship. Implications for educational practice are also presented.

### 4.1. The Contribution of Task Value to the Content Specificity and Generality of the Relationship

One of this study's noteworthy findings is that students that perceived a higher task utility value of learning mathematics for university entrance examinations at both the end of the first and second terms had higher cognitive engagement and, therefore, were more able to solve mathematical problems. In comparison, students that perceived a higher task utility value of learning mathematics for university entrance examinations at the end of either the first or second terms had higher math evaluation anxiety. These findings suggest that perceiving that learning mathematics has utility value for university entrance examination enhances mathematical problem solving, which is an adaptive outcome in high school mathematics and has the side effect of increasing math evaluation anxiety. These findings are consistent with those of previous studies that have shown that mathematics utility value is positively associated with problem solving [5] but contradict the present study's assumption that high mathematics task value is associated with adaptive emotion.

An explanation for these findings is that the utility value for entrance examinations is a construct similar to extrinsic motivation in self-determination theory [15]. In organic integration theory [39], one of the sub-theories of self-determination theory, extrinsic motivation is subdivided into four levels according to the degree of self-determination: external regulation, introjected regulation, identified regulation, and integrated regulation. A higher degree of self-determination is positively associated with achievement and adaptive to achievement and negatively associated with negative emotions such as anxiety [40]. It would seem that perceiving that learning math has a high utility value for university entrance examinations at the ends of both the first and second terms functioned as a high degree of self-determination in the identified regulation or integrated regulation levels, while this perception at either the first or second term-end functioned as a low degree of self-determination in external regulation. This finding suggests that it would be effective to consistently promote the utility value of learning mathematics for entrance examinations across various contents to improve mathematical problem solving. On the other hand, a temporary increase in mathematics' utility value for entrance examinations only when students are learning certain content should be avoided since it may increase math assessment anxiety.

Another notable finding of this study is that interest value was associated directly with math anxiety and engagement and indirectly with mathematical problem solving via emotional and cognitive engagement in both the between and within models (Figure 2), while the absolute value of the path coefficient was greater in the between model (Table 5). These results not only support the related processes hypothesized by this study (Hypotheses 1 and 4) in both the within and between models but also suggest strong content generality in the processes. Therefore, to reduce math anxiety and improve engagement and mathematical problem solving, increasing interest value consistently across different contents may be more effective, rather than temporarily increasing it only when students learn certain content.

When statistically controlling for interest value, students with higher attainment value at either the end of the first or second terms were more anxious about learning mathematics. This result contradicts the present study's assumption that high task value in mathematics is associated with achievement and adaptive emotions (Hypotheses 1). It would appear that attainment value tends to be related to ego-involvement when interest value is statistically controlled for since it can be interpreted as the importance of doing the task well, excluding

the enjoyment and fun of doing the task. In other words, achievement value that does not overlap with interest value is considered similar to performance goals in achievement goal theory. Since performance goals are positively associated with negative emotions such as anxiety [41], it can be assumed that attainment value was positively associated with math learning anxiety in this study. In light of the above, high attainment value only in certain content should be avoided as it may induce math learning anxiety.

### 4.2. The Contribution of Math Anxiety to the Content Specificity and Generality of the Relationship

Math anxiety was not significantly associated with engagement or mathematical problem solving in either the between or within models. This result is inconsistent with the assumption of the present study and of the control-value theory [21] that math anxiety is negatively associated with engagement and mathematical problem solving (Hypotheses 2 and 5). However, when modelled as a single predictor, as shown in Table 2, mathematics anxiety was negatively associated with engagement and mathematical problem solving. Therefore, it may be assumed that math anxiety has a more minor influence on mathematical problem solving than task value and engagement and a more minor influence on engagement than task value, which is why no association was found in the multilevel structural equation modelling. These findings suggest that interventions that focus on enhancing the perceived value learning mathematics, rather than alleviating math anxiety, would be more effective for improving engagement and mathematical problem solving.

Another explanation for the lack of a link between math anxiety and engagement is that learning engagement is a concept that conflicts with but does not directly reflect math avoidance. Disaffection is a concept that is the opposite of engagement, such as withdrawing from learning [25,34]. Thus, math anxiety is more likely to be directly related to disaffection rather than engagement in both within and between models.

### 4.3. The Contribution of Engagement to the Content Specificity and Generality of the Relationship

In both the between and within models, the results showed that students with higher emotional engagement had higher cognitive engagement and were, therefore, better able to solve mathematical problems. This result supports the assumption that emotions have motivational aspects that direct cognition [29] and that cognitive engagement, among other forms of engagement, is directly and strongly related to achievement [26] (Hypotheses 3). As Fung et al. [26] highlighted, cognitive engagement in learning involves various aspects and is, at the same time, an indicator of learning motivation, which is why the present study found it to be significantly associated with mathematical problem solving.

An important finding is that the standardized path coefficients from cognitive engagement to mathematical problem solving were larger in the between model than in the within model. This suggests that a consistent increase in cognitive engagement is more effective in improving mathematical problem solving than a temporary increase that occurs only when students learn certain content. On the other hand, the standardized path coefficients from emotional to cognitive engagement were similar in the within and between models, suggesting that both an increase in emotional engagement that occurs only when students learn certain content, as well as consistent increases, may improve cognitive engagement.

In addition, behavioral engagement was not significantly associated with mathematical problem solving in both the between and within models, which is inconsistent with behavioral engagement being associated with mathematical achievement [11]. However, as shown in Table 3, behavioral engagement was positively correlated with mathematical problem solving when modelled as a single predictor. Therefore, it can be assumed that behavioral engagement contributes less to mathematical problem solving than cognitive engagement and, consequently, was not found to be associated in the multilevel structural equation modelling. These findings suggest that cognitive engagement, such as planning and monitoring learning and learning strategies, is more effective in improving mathematical problem solving than involvement in learning, for example, learners' attitudes toward the task or the effort and persistence they demonstrate when undertaking it.

*4.4. Limitations*

This study had four limitations. First, it is necessary to consider the strong predictors of the mathematical problem solving component that are unique to each content type because the variance explained by mathematical problem solving in the within model was slight (3%). For example, self-efficacy may be a stronger predictor of mathematical problem solving since it is a critical component of expectations in the expectancy-value theory [15] and is positively associated with mathematics performance after statistically controlling for sociocultural circumstances and personality [42]. Second, it is unclear to what extent the study's findings can be generalized to other mathematical problem solving measures since the only measure of problem solving used was School A's test scores. Thus, future studies should investigate other measures besides test scores. Third, caution must be applied since the study only used vectors and matrices as learning contents; therefore, future research should be conducted with different learning content. Fourth, the validity of the task value, math anxiety and engagement scales in this study are not sure to be sufficient due to the small number of items and the self-reporting nature of the study. The number of items needed to be reduced in this study as it was impossible to overburden the study participants. Furthermore, the engagement scale in this study was on a different Likert scale from previous studies [34,35]. Future research needs to use all Likert scale items that have been validated in previous studies and a combination of self-report and teacher-report.

**Funding:** This research received no external funding.

**Institutional Review Board Statement:** The study was conducted according to the guidelines of the Declaration of Helsinki. No ethical review or approval was required for this study, in accordance with the local legislation and institutional requirements because the survey was conducted as part of the improvement of math education practice in School A, written informed consent was obtained from all subjects in advance, an anonymous survey in which individuals were not identified, and no research invasion occurred. In Japan, ethical review is not necessarily required for this type of anonymous and non-invasive educational research.

**Informed Consent Statement:** Written informed consent was obtained from all students involved in the study.

**Data Availability Statement:** The datasets of the current study are not publicly available. However, data of the current study will be available from the corresponding author on reasonable request with permission.

**Conflicts of Interest:** The author declares no conflict of interest.

## Appendix A. Items of Task Value, Math Anxiety and Engagement

*Appendix A.1. Task Value*

Utility value for practice

- Mathematics is useful in my daily life.
- Learning mathematics helps me to understand how events and phenomena around me function.
- Knowing the content of mathematics well is useful in everyday life.
- Utility value for entrance examinations
- Learning mathematics is important for entrance exams.
- Learning mathematics is important for me to achieve my desired pathway.
- Learning mathematics will help me to get a desirable career.

Interest value

- The content of mathematics is interesting.
- The content of mathematics is fun.
- The content of mathematics is curious.

Attainment value

- Understanding what you learn in mathematics can help me grow.
- Learning mathematics will bring me closer to my 'ideal self'.
- People who know mathematics well are smart.

*Appendix A.2. Math Anxiety*

Math learning anxiety

- When I take a math class, I am . . .
- When I am waiting for my math teacher to arrive before class, I am . . .
- When I solve math problems, I am . . .
- When I work on my math homework, I am . . .

Math evaluation anxiety

- When I think about tomorrow's math test, I am . . . .
- When I take my math test, I am . . . .
- When I take my math final exam, I am . . . .
- When I think about my math exam, I am . . . .

*Appendix A.3. Learning Engagement*

Behavioral engagement

- I try hard to learn mathematics.
- I work as hard as I can to learn mathematics.
- I pay attention to learning mathematics.

Emotional engagement

- I enjoy learning mathematics.
- When I learn mathematics, I feel good.
- When I learn mathematics, I feel interested.

Cognitive engagement

- I try to connect what I am learning with my knowledge.
- As I learn mathematics, I keep track of how much I understand.
- When I learn mathematics, I try to understand, not just to get the correct answer.

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
