# Peer review of "The Content Specificity and Generality of the Relationship between Mathematical Problem Solving and Affective Factors"

_psych, doi:10.3390/psych4030044_

Round 1

Reviewer 1 Report

This study looks at multiple paths between task values (utility, attainment, ad interest) on mathematical problem-solving (the scores on the maths exams). Overall, I found this a clear study that was well-contextualized within the literature. My only comments relate to descriptions and expositions:

1. The description of within and between models in lines 212-215 needs to be clearer. Importantly, the within and between models in table 4 are not the same. How should we interpret this, especially as the indirect effects in table 5 show a different pattern?
2. Can you indicate in table 2 which correlations are significant?
3. Can you create a figure like figure 1 for table 4 and include the beta coefficients on the paths?
4. The hypothesis was that math anxiety was related to engagement (H2) and indirectly to problem-solving (H5). Neither survived the SEM. Is this pattern due to the lack of having measures of 'math avoidance', as hinted in lines 87-100? Relatedly, given that section 1.2 is almost entirely about the impact of math avoidance, why was there no measurement attempted?

Author Response

I would like to thank reviewer#1 for providing valuable suggestions and insightful comments.

COMMENT#1: 1. The description of within and between models in lines 212-215 needs to be clearer. Importantly, the within and between models in table 4 are not the same. How should we interpret this, especially as the indirect effects in table 5 show a different pattern?

RESPONSE#1: Thank you for pointing this out. I made the following additions and revisions.

First, I added the following explanation for the description of within and between models.

  • The data in this study are two-level hierarchical data (Line 238-241)
  • The primary extraction unit in this study (Line 241-243)
  • Meaning and examples of within and between model (Line 243-253)

Second, I made the following revisions to my discussion of the differences between within model and between models. Incidentally, the main finding of this study, the difference in utility value for the entrance examinations, is discussed in Lines 358-387

  • strong content generality in the relationship between interest value, engagement, and mathematical problem-solving (Line 388-397)
  • strong content generality in the relationship between cognitive engagement and mathematical problem-solving (Line 439-443)

Third, I corrected the error in Table 5.

COMMENT#2: 2. Can you indicate in table 2 which correlations are significant?

RESPONSE#2: Revised as suggested (Table. 3, Line 285-286).

COMMENT#3: 3. Can you create a figure like figure 1 for table 4 and include the beta coefficients on the paths?

RESPONSE#3: Revised as suggested (Figure .2, Line 307-313).

COMMENT#4: 4. The hypothesis was that math anxiety was related to engagement (H2) and indirectly to problem-solving (H5). Neither survived the SEM. Is this pattern due to the lack of having measures of 'math avoidance', as hinted in lines 87-100? Relatedly, given that section 1.2 is almost entirely about the impact of math avoidance, why was there no measurement attempted?

RESPONSE#4: I did not measure it in this study because I considered engagement to be a counter-concept to math avoidance. Based on your point, hypothesis 2 may not have been supported as it did not directly measure math avoidance. Thus, I added references to engagement, math avoidance and math anxiety in the discussion (Line 425-429).

Reviewer 2 Report

This is a well-written and analyzed research paper. However, my concern is that the data may not support the intended variables well. The scales were all rather short, some only three items in length, and the whole self-report data set, though presented as several scales would have comprised a single questionnaire. They seemed to be ‘adapted from’ other scales, and also seem to represent abridged versions of the original scales. While I appreciate that the author provided reliability data, I have many doubts that what was measured corresponds well to the intended traits. There is a similar issue with using test scores as a measure of cognitive ability. The author could perhaps rebut this with additional scale information or offer additional information on the validity of the measures.

Another important issue is that on Line 154, the author describes obtaining the consent of the teacher, but did they take informed consent from the participants? The research probably should have been approved by a research ethics committee, and that would usually involve taking written informed consent from all participants. The ethics committee possibly could have waived that. But if feels to me that as not only questionnaire data was collected, but also school grades, individual consent would likely have been required, in fact parental consent would have been required. Details on ethical review would be useful here.

Minor issues:

Line 124. A previous study is mentioned, that should be cited. If it is an unpublished work, it may be best not to refer to it as a previous study, rather, this is based on your research into the topic.

Line 141, it would be best if age and sex information about the participants is provided, if available. At present most reader would have little idea of the age of the participants. Also, more details of the sample recruitment is needed, for example did potential recruits decline to participate, were some cases excluded due to incomplete data, spoiled questionnaires etc.?

 Table 2: Which correlation type (e.g., Pearson, Spearman) was used?

Author Response

I would like to thank reviewer#1 for providing valuable suggestions and insightful comments.

COMMENT#1: This is a well-written and analyzed research paper. However, my concern is that the data may not support the intended variables well. The scales were all rather short, some only three items in length, and the whole self-report data set, though presented as several scales would have comprised a single questionnaire. They seemed to be ‘adapted from’ other scales, and also seem to represent abridged versions of the original scales. While I appreciate that the author provided reliability data, I have many doubts that what was measured corresponds well to the intended traits. There is a similar issue with using test scores as a measure of cognitive ability. The author could perhaps rebut this with additional scale information or offer additional information on the validity of the measures.

RESPONSE#1: Thank you for pointing this out. I added the following explanations as to why the scale used in this study is considered to have validity.

  • The validity of the original scales adopted and modified in this study (Task value, Line 171-174; Math anxiety, Line 185-186; Engagement, Line 195-197)
  • The mathematical problem-solving scale in this study was appropriate in terms of content and number of problems (Line 206-213)
  • Results of reliability and validity testing of the scales used in this study (Analysis procedure, Line 215-225; Results, Line 258-281)

The scales used in this study were reliable and valid, but increasing the number of items and using methods other than self-report. I added Line 472-477 about this.

COMMENT#2: Another important issue is that on Line 154, the author describes obtaining the consent of the teacher, but did they take informed consent from the participants? The research probably should have been approved by a research ethics committee, and that would usually involve taking written informed consent from all participants. The ethics committee possibly could have waived that. But if feels to me that as not only questionnaire data was collected, but also school grades, individual consent would likely have been required, in fact parental consent would have been required. Details on ethical review would be useful here.

RESPONSE#2: As you pointed out, I added the following information.

  • Method of informing the public about the purpose and content of the survey (Line 148-150)
  • Reasons for waiving ethical review (Line 478-483)
  • Informed consent obtained from all the participants in the study (Line 484-485)

COMMENT#3: Line 124. A previous study is mentioned, that should be cited. If it is an unpublished work, it may be best not to refer to it as a previous study, rather, this is based on your research into the topic.

RESPONSE#3: Revised as suggested (Line 124).

COMMENT#4: Line 141, it would be best if age and sex information about the participants is provided, if available. At present most reader would have little idea of the age of the participants. Also, more details of the sample recruitment is needed, for example did potential recruits decline to participate, were some cases excluded due to incomplete data, spoiled questionnaires etc.?

RESPONSE#4: I added the following information that I got. However, I could not obtain data on the age of the participants.

  • Sex and ethnicity of participants (Line 141, 143)
  • The sample recruitment (Line 144-150)
  • Reason for missing data (Line 162-166)
  • Methods for dealing with missing values (Line 255-256)

COMMENT#5: Table 2: Which correlation type (e.g., Pearson, Spearman) was used?

RESPONSE#5: I used Pearson correlation (Table 3).

Round 2

Reviewer 2 Report

Some recruitment and demographic information were included which improves the manuscript. Even if individual ages were not collected, the author could perhaps say what a typical age would be for students in that grade of the Japanese school system.

I feel that there are still some problems with the scales. The Japanese version of the math anxiety scale that is cited is to a 34-item version, the version used in this study was 8 items. I appreciate that it is mentioned that it is based on a different version, but the fact remains that a non-standard questionnaire was used. The validity information has the same citation, I assume that it refers to the 34-item version. Similarly, Skinner’s scale of Behavioral and Emotional Engagement includes 10 items, and your version included 6 of them, plus 3 from a different scale. Scores were given on a 6-point Likert scale, though the original used a 4-point scale. The validity information is to a scale that is ‘similar’ to the one that the authors used. That doesn’t really show validity.  There are similar issues with the other scales. For this reason, I feel that there are substantial doubts concerning the validity of the measures. The author’s explanation about why the abridged versions are valid and reliable , apparently on lines 472-476, doesn’t deal with that issue.

Regarding the lack of ethics committee review, I still have concerns about this. It is now said in the manuscript, that the need for ethical review was waived (lines 529-534). But notably, that is the passive voice. It is not clear whether an ethics committee waived the need for review, or whether the investigators waived it for themselves. Informed consent was taken, but notably not written consent. It seems to me that school children would be considered vulnerable participants, and that the protocol should have been reviewed by, or granted exception from review, by a recognized research ethics committee.

Author Response

I would like to thank reviewer 2 for providing valuable suggestions and insightful comments.

Point 1: Some recruitment and demographic information were included which improves the manuscript. Even if individual ages were not collected, the author could perhaps say what a typical age would be for students in that grade of the Japanese school system.

Response 1: Thank you for pointing this out. I added the typical age of Japanese high school second-year students (Line 145-146).

Point 2: I feel that there are still some problems with the scales. The Japanese version of the math anxiety scale that is cited is to a 34-item version, the version used in this study was 8 items. I appreciate that it is mentioned that it is based on a different version, but the fact remains that a non-standard questionnaire was used. The validity information has the same citation, I assume that it refers to the 34-item version. Similarly, Skinner’s scale of Behavioral and Emotional Engagement includes 10 items, and your version included 6 of them, plus 3 from a different scale. Scores were given on a 6-point Likert scale, though the original used a 4-point scale. The validity information is to a scale that is ‘similar’ to the one that the authors used. That doesn’t really show validity.  There are similar issues with the other scales. For this reason, I feel that there are substantial doubts concerning the validity of the measures. The author’s explanation about why the abridged versions are valid and reliable , apparently on lines 472-476, doesn’t deal with that issue.

Response 2: Thank you for pointing this out. Learning from your suggestions, I deleted the following statements.

  • It is confirmed that the Japanese Mathematics Anxiety Scale [32] satisfies adequate convergent and discriminant validity and reliability.
  • Shimizu's [27] engagement scale, similar to the present study, has adequate convergent and discriminant validity and reliability and is positively correlated with mathematical problem-solving.

In addition, as you pointed out, I added the following information.

  • The author and two mathematics teachers from junior high and high schools confirmed the content of items of task value, math anxiety, and engagement (Line 171-174).
  • The Likert scale of the engagement scale in this study differs from previous studies and the reasons for this difference (Line 198-201).
  • Limitation related to the validity of the engagement scale (Line 477-480).

Point 3: Regarding the lack of ethics committee review, I still have concerns about this. It is now said in the manuscript, that the need for ethical review was waived (lines 529-534). But notably, that is the passive voice. It is not clear whether an ethics committee waived the need for review, or whether the investigators waived it for themselves. Informed consent was taken, but notably not written consent. It seems to me that school children would be considered vulnerable participants, and that the protocol should have been reviewed by, or granted exception from review, by a recognized research ethics committee.

Response 3: Thank you for pointing this out. No ethical review or approval was required for this study, in accordance with the local legislation and institutional requirements because the survey was conducted as part of the improvement of math education practice in School A, written informed consent was obtained from all subjects in advance, an anonymous survey in which individuals were not identified, and no research invasion occurred (Line 481-487). Therefore, the author waived the ethical review in accordance with the protocols for ethical review in Japan (URL: https://www.mhlw.go.jp/stf/seisakunitsuite/bunya/hokabunya/kenkyujigyou/i-kenkyu/index.html ).

Furthermore, written informed consent was obtained from all students (Line 488-489).
